

# A novel hybrid-based approach of snort automatic rule generator and security event correlation (SARG-SEC)

Ebrima Jaw[1,2] and Xueming Wang[1]

[1] College of Computer Science and Technology, Guizhou University, Guiyang, Guizhou, China
[2] School of Information Technology and Communication, University of The Gambia (UTG), Banjul, Peace Building, Kanifing, The Gambia

## ABSTRACT

The rapid advanced technological development alongside the Internet with its cutting-edge applications has positively impacted human society in many aspects. Nevertheless, it equally comes with the escalating privacy and critical cybersecurity concerns that can lead to catastrophic consequences, such as overwhelming the current network security frameworks. Consequently, both the industry and academia have been tirelessly harnessing various approaches to design, implement and deploy intrusion detection systems (IDSs) with event correlation frameworks to help mitigate some of these contemporary challenges. There are two common types of IDS: signature and anomaly-based IDS. Signature-based IDS, specifically, Snort works on the concepts of rules. However, the conventional way of creating Snort rules can be very costly and error-prone. Also, the massively generated alerts from heterogeneous anomaly-based IDSs is a significant research challenge yet to be addressed. Therefore, this paper proposed a novel Snort Automatic Rule Generator (SARG) that exploits the network packet contents to automatically generate efficient and reliable Snort rules with less human intervention. Furthermore, we evaluated the effectiveness and reliability of the generated Snort rules, which produced promising results. In addition, this paper proposed a novel Security Event Correlator (SEC) that effectively accepts raw events (alerts) without prior knowledge and produces a much more manageable set of alerts for easy analysis and interpretation. As a result, alleviating the massive false alarm rate (FAR) challenges of existing IDSs. Lastly, we have performed a series of experiments to test the proposed systems. It is evident from the experimental results that SARG-SEC has demonstrated impressive performance and could significantly mitigate the existing challenges of dealing with the vast generated alerts and the labor-intensive creation of Snort rules.

# INTRODUCTION

The advent of the Internet has come with the cost of wide-scale adoption of innovative technologies such as cloud computing (*Al-Issa, Ottom & Tamrawi, 2019*), artificial intelligence (*Miller, 2019*), the Internet of Things (IoT) (*Park, 2019*), and vast ranges of web-based applications. Therefore, leading to considerable security and privacy challenges of

Corresponding author
Xueming Wang,
xmwang3@gzu.edu.cn

managing these cutting-edge applications using traditional security and privacy protection mechanisms such as firewall, anti-virus, virtual private networks (VPNs), and anti-spyware (*Meryem & Ouahidi, 2020*). However, due to the vast range of competitive solutions such as higher efficiencies, scalability, reduced costs, computing power, and most importantly, the delivery of services, these technologies continue to revolutionize various aspects of our daily lives drastically, for instance, in the health care systems, research industry, the global business landscape, government, and private sectors (*Xue & Xin, 2016*).

Moreover, countless types of cyber-attacks have evolved dramatically since the inception of the Internet and the swift growth of ground-breaking technologies. For example, social engineering or phishing (*Kushwaha, Buckchash & Raman, 2017*), zero-day attack (*Jyothsna & Prasad, 2019*), malware attack (*McIntosh et al., 2019*), denial of service (DoS) (*Verma & Ranga, 2020*), unauthorized access of confidential and valuable resources (*Saleh, Talaat & Labib, 2019*). Additionally, according to the authors of *Papastergiou, Mouratidis & Kalogeraki (2020)*, a nation's competitive edge in the global market and national security is currently driven by harnessing these efficient, productive, and highly secure leading-edge technologies with intelligent and dynamic means of timely detection and prevention of cyberattacks. Nevertheless, irrespective of the tireless efforts of security experts in defense mechanisms, hackers have always found ways to get away with targeted resources from valuable and most trusted sources worldwide by launching versatile, sophisticated, and automated cyber-attacks. As a result, causing tremendous havoc to governments, businesses, and even individuals (*Sarker et al., 2020*).

For instance, the authors of *Damaševičius et al. (2021)* intriguingly review various cyber-attacks and their consequences. Firstly, the paper highlights the estimated 6 trillion USD of cyber-crimes by 2021 and the diverse global ground-breaking cyber-crimes that could lead to the worldwide loss of 1 billion USD. Finally, it highlights a whopping 1.5 trillion USD of cyberattack revenues resulting from two to five million computers compromised daily. Furthermore, according to published statistics of AV-TEST Institute in Germany, during the year 2019, there were more than 900 million malicious executables identified among the security community, and data breach costing 8.19 million USD for the United States, predicted to grow in subsequent years. Moreover, the Congressional Research Service of the USA has highlighted that cybercrime-related incidents have cost the global economy an annual loss of 400 billion USD.

Similarly, 2016 alone recorded more than a whopping 3 billion zero-day attacks and approximately 9 billion stolen data records since 2013 (*Khraisat et al., 2019*). In addition, the energy sector in Ukraine suffered catastrophic coordinated cyberattacks (APT) that led to a significant blackout affecting more than 225,000 people. They also highlighted similar alarming APT threats, such as DragonFly, TRITON, and Crashoverride, that could cause devasting consequences to individual lives and the global economy, thereby leading to national security threats (*Grammatikis et al., 2021*). Accordingly, it is essential for security experts to design, implement and deploy robust and efficient cybersecurity frameworks to alleviate the current and subsequent alarming losses for the government and private sectors. Additionally, it is an urgent and crucial challenge to effectively identify the increasing cyber incidents and cautiously protect these relevant applications from such cybercrimes.

Therefore, the last few decades have witnessed Intrusion Detection Systems (IDSs) increasing in popularity due to their inherent ability to detect an intrusion or malicious activities in real-time. Consequently, making IDSs critical applications to safeguard numerous networks from malicious activities (*Dang, 2019*; *Meryem & Ouahidi, 2020*). Finally, James P. Anderson claimed credit for the inception of the IDS concept in his paper written in 1980 (*Anderson, 1980*), highlighting various methods of enhancing computer security threat monitoring and surveillance.

Intrusion Detection is the procedure of monitoring the events occurring in a computer system or network and analyzing them for signs of intrusion", similarly, an intrusion is an attempt to bypass the security mechanisms of a network or a computer system, thereby compromising the Confidentiality, Integrity, and Availability (CIA) (*Kagara & Md Siraj, 2020*). Moreover, an IDS is any piece of hardware or software program that monitors diverse malicious activities within computer systems and networks based on network packets, network flow, system logs, and rootkit analysis (*Bhosale, Nenova & Iliev, 2020*; *Liu & Lang, 2019*). Misused detection (knowledge or signature-based) and anomaly-based methods are the two main approaches to detecting intrusions within computer systems or networks. Nevertheless, the past decade has witnessed the rapid rise of the hybrid-based technique, which typically exploits the advantages of the two methods mentioned above to yield a more robust and effective system (*Saleh, Talaat & Labib, 2019*).

Misused IDS (MIDS) is a technique where specific signatures of well-known attacks are stored and eventually mapped with real-time network events to detect an intrusion or intrusive activities. The MIDS technique is reliable and effective and usually gives excellent detection accuracy, particularly for previously known intrusions. Nevertheless, this approach is questionable due to its inability to detect novel attacks. Also, it requires more time to analyze and process the massive volume of data in the signature databases (*Khraisat et al., 2020*; *Lyu et al., 2021*). The authors of *Jabbar & Aluvalu (2018)* presented an exceptional high-level SIDS architecture, which includes both distributed and centralized modules that effectively enhanced the protection of IoT networks against internal and external threats. Furthermore, the authors exploit the Cooja simulator to implement a DoS attack scenario on IoT devices that rely on version number modification and "Hello" flooding. Finally, the authors claimed that these attacks might influence specific IoT devices' reachability and power consumption.

In contrast, the anomaly-based detection method relies on a predefined network behavior as the crucial parameter for identifying anomalies and commonly operates on statistically substantial network packets. For instance, incoming network packets or transactions are accepted within the predefined network behavior. Otherwise, the anomaly detection system triggers an alert of anomaly (*Kagara & Md Siraj, 2020*). It is essential to note that the main design idea of the anomaly detection method is to outline and represent the usual and expected standard behavior profile through observing activities and then defining anomalous activities by their degree of deviation from the expected behavior profile using statistical-based, knowledge-based, and machine learning-based methods (*Jyothsna & Prasad, 2019*; *Khraisat et al., 2020*). The acceptable network behavior can be learned using the predefined network conditions, more like blocklists or allowlists that determine the

network behavior outside a predefined acceptable range. For instance, "detect or trigger an alert if ICMP traffic becomes greater than 10% of network traffic" when it is regularly only 8%.

Finally, the anomaly-based approach provides a broader range of advantages such as solid generalizability, the ability to determine internal malicious activities, and a higher detection rate of new attacks such as the zero-day attack. Nevertheless, the most profound challenge is the need for these predefined baselines and the substantial number of false alarm rates resulting from the fluctuating cyber-attack landscape (*Einy, Oz & Navaei, 2021*). For instance, (*Fitni & Ramli, 2020*) intelligently used logistic regression, decision tree, and gradient boosting to propose an optimized and effective anomaly-based ensemble classifier. The authors claimed impressive findings such as 98.8% performance accuracy, 98.8%, 97.1%, and 97.9% precision, recall, and F1-score.

The hybrid-based intrusion detection systems (HBIDS) exploit the functionality of MIDS to detect well-known attacks and flag novel attacks using the anomaly method. High detection rate, accuracy, and fewer false alarm rates are some of the main advantages of this approach (*Khraisat et al., 2020*). The authors of *Khraisat et al. (2020)* suggested an efficient and lightweight hybrid-based IDS that mitigates the security vulnerabilities of the Internet of Energy (IoE) within an acceptable time frame. The authors intelligently exploit the combined strengths of K-means and SVM and utilize the centroids of K-means to enhance the process of training and testing the SVM model. Moreover, they selected the best value of "k" and fine-tuned the SVM for best anomaly detection and claimed to have drastically reduced the overall detection time and impressive performance accuracy of 99.9% compared to current cutting-edge approaches.

Additionally, the two classical IDS implementation methods are Network Intrusion Detection Systems (NIDS) (*Mirsky et al., 2018*) and Host-based IDS (HIDS) (*Aung & Min, 2018*). A HIDS detection method monitors and detects internal attacks using the data from audit sources and host systems like firewall logs, database logs, application system audits, window server logs, and operating systems (*Khraisat et al., 2019*). In contrast, NIDS is an intrusion detection approach that analyses and monitors the entire traffic of computer systems or networks based on flow or packet-based and tries to detect and report anomalies. For example, the distributed denial of service (DDoS), denial of service (DoS), and other suspicious activities like internal illegal access or external attacks (*Niyaz et al., 2015*). Unlike HIDS, NIDS usually protects an entire network from internal or external intrusions. However, such a process can be very time-consuming, high computational cost, and very inefficient, especially in most current cutting-edge technologies with high-speed communication systems.

Nevertheless, this approach still has numerous advantages (*Bhuyan, Bhattacharyya & Kalita, 2014*). For instance, it is more resistant to attacks compared to HIDS. Furthermore, it monitors and analyses the complete network's traffic if appropriately located in a network, leading to a high probability of detection rate. Also, NIDS is platform-independent, thus enabling them to work on any platform without requiring much modification. Finally, NIDS does not add any overhead to the network traffic (*Othman et al., 2018*).

Snort is a classic example of NIDS (*Sagala, 2015*). Irrespective of the availability of other signature-based NIDS, such as Suricata and Zeek (Bro) (*Ali, Shah & Issac, 2018*), this work adopted Snort because it is the leading open-source NIDS with active and excellent community support. Likewise, it is easy to install and run with readily available online resources. Furthermore, because Suricata can use Snort's rulesets but is prone to false positives with the need for an intensive system and network resource, we concluded to use Snort for the proposed solutions.

Snort is a classical open-source NIDS that has the unique competence of performing packet logging and real-time network traffic analysis within computer systems and networks using content searching, matching, and protocol analysis (*Aickelin, Twycross & Hesketh-Roberts, 2007*; *Tasneem, Kumar & Sharma, 2018*). Therefore, significantly contributing to the protection of some major commercial networks. Snort is a signature-based IDS that detects malicious live Internet or network traffic utilizing the predefined Snort rules, commonly applied in units of packets' header, statistical information (packet size), and payload information. Thus, it has a unique feature of high detection rate and accuracy (*Tasneem, Kumar & Sharma, 2018*). However, it cannot detect novel attacks and, at the same time, it requires expert knowledge to create and update rules frequently, which is both costly and faulty.

Similarly, IDS have emerged as popular security frameworks that significantly minimized various cutting-edge cyber-attacks over the past decade. Anomaly-based IDS has gained quite a buzz among network and system administrators for monitoring and protecting their networks against malicious attempts, which has achieved phenomenal success, especially in detecting and protecting systems and networks against novel or zero-day attacks (*Tama, Comuzzi & Rhee, 2019*). However, it comes with costly negative consequences of generating thousands or millions of false-positive alerts or colossal amounts of complex data for humans to process and make timely decisions (*Sekharan & Kandasamy, 2018*). As a result, administrators ignore these massive alerts, which creates room for potential malicious attacks against highly valued and sensitive information within a given system or network.

Accordingly, the past years have seen a growing interest in designing and developing network security management frameworks from academia and industry, which involves analyzing and managing the vast amount of data from heterogeneous devices, commonly referred to as event correlation. Event correlation has significantly mitigated modern cyber-attacks challenges using its unique functionality of efficiently and effectively analyzing and making timely decisions from massive heterogeneous data (*Suarez-Tangil et al., 2009*). Consequently, the past years have seen many researchers and professionals exploit the efficiency of event correlation techniques to address the problems mentioned earlier (*Dwivedi & Tripathi, 2015*; *Ferebee et al., 2011*; *Guillermo Suarez-Tangil et al., 2015*).

Nevertheless, this field of research is at its infant stage as minimal work is done to address these issues. Likewise, according to the authors' knowledge, none of the above solutions provides comprehensive solutions that address the manual creation of Snort rules and the event correlation as a single solution. Based on the above challenges, this paper proposed two effective and efficient approaches to address the problems associated with the manual

creation of Snort rules and mitigating the excessive false alarm rates generated by current IDSs. First, we present an automatic rule creation technique that focuses on packet header and payload information. Generally, we need to find standard features by examining all the network traffic to create a rule. Nonetheless, this method is inefficient and requires much time to complete the rule, and the accuracy of the rules made is variable according to the interpreters' ability. Therefore, various automatic rule (signature) generation methods have been proposed (*Sagala, 2015*).

However, most of these methods are used between two specific strings, which is still challenging for creating reliable and effective Snort rules. Therefore, we present a promising algorithm based on the content rules, enabling the automatic and easy creation of Snort rules using packet contents. Secondly, we equally proposed a novel model that efficiently and effectively correlates and prioritizes IDS alerts based on the severity using various features of a network packet. Moreover, the proposed system does not need prior knowledge while comparing two different alerts to measure the similarity in diverse attacks. The following are an overview of the main contributions of this research work:

○ The authors proposed an optimized and efficient Snort Automatic Rule Generator (SARG) that automatically generates reliable Snort rules based on content features.
○ Similarly, we present a novel Security Event (alert) Correlator (SEC) that drastically and effectively minimized the number of alerts received for convenient interpretation.
○ This paper also provides solid theoretical background knowledge for the readership of the journal to clearly understand the fundamental functions and capabilities of Snort and various correlation methods.
○ Finally, the proposed approach has recorded an acceptable number of alerts, which directly correlates with significantly mitigating the challenges of false alarm rates.

The rest of the paper is organized as follows: 'Essential Concepts' discusses important background concepts. Similarly, 'Materials and Methods' explains the materials and methods of the proposed system. Next, 'Results and Discussion' highlights the results and discussions of the proposed approach. Finally, the paper concludes in 'Conclusions'.

## ESSENTIAL CONCEPTS

This section presents brief essential concepts that support the work in this research paper, which will provide readers with the necessary knowledge to appreciate this research and similar results better.

### Synopsis of HIDS and NIDS functionalities

The crucial advantages of a host-based IDS are; the ability to detect internal malicious attempts that might elude a NIDS, the freedom to access already decrypted data compared to NIDS, and the ability to monitor and detect advanced persistent threats (APT). In contrast, some of its disadvantages are; firstly, they are expensive as it requires lots of management efforts to mount, configure and manage. It is also vulnerable to specific DoS attacks and uses many storage resources to retain audit records to function correctly (*Liu et al., 2019*; *Saxena, Sinha & Shukla, 2017*). The authors of *Arrington et al. (2016)* use the innovative strength of machine learning such as artificial immune systems to present an

interesting host-based IDS. Finally, they claimed to have achieved a reasonable detection rate through rescinding out the noise within the environment.

The central idea of any classical NIDS is using rulesets to identify and alert malicious attempts. The majority of the NIDS comes with pre-installed rules that can be modified to target specific attacks (*Ojugo et al., 2012*). For example, creating a rule for a possible probing attack and saving it in the *local.rules* of Snort IDS will ensure an alert is raised whenever an intrusion is initiated that matches the rule. Furthermore, the essential functionalities of a classical NIDS are: practical identification and alerting of policy violations, suspicious unknown sources, destination network traffics, port scanning, and other common malicious attempts. However, the bulk of NIDS requires costly hardware with expensive enterprise solutions, making it hard to acquire (*Elrawy, Awad & Hamed, 2018*). The authors of (*Nyasore et al., 2020*) presented an intriguing challenge of evaluating the overlap among various rules in two rulesets of the Snort NIDS. However, the work failed to assess the distinction between diverse rulesets explicitly.

Consequently, the work presented in *Sommestad, Holm & Steinvall (2021)* provides an interesting empirical analysis of the detection likelihood of 12 Snort rulesets against 1143 misuse attempts to evaluate their effectiveness on a signature-based IDS. Similarly, they listed certain features as the determining factor of the detection probability. Finally, they claimed impressive results such as a significant 39% raise of priority-1-alerts against the misuse attempts and 69–92% performance accuracy for various rulesets.

Figure 1 illustrates a standard representation of a HIDS and NIDS architecture with unique functionalities in detecting malicious activities. For instance, Fig. 1 shows a malicious user (attacker) who initiated a DDoS attack against one of the internal servers within the LAN. However, due to the internal security mechanisms, packets are inspected by the firewall as the first layer of protection. Interestingly, some malicious packets can bypass the firewall due to the cutting-edge attack mechanisms, necessitating NIDS (*Bul'ajoul, James & Pannu, 2015*). Therefore, the NIDS receives the packets and does a further packet inspection. If there are any malicious activities, the packets are blocked and returned to the firewall. Then, the firewall will drop the packets or notify the network administrator, depending on the implemented policies. It is crucial to note that the same applies to the outgoing packets from the LAN to the WAN.

In contrast, the scenario depicted at the top of Fig. 1 shows a regular user requesting web services. Initially, the request goes *via* the same process as explained above. Then, if the NIDS qualifies the requests, it is sent to the server through the network switch to the webserver. Furthermore, the server responds with the required services, which goes through the same process as the reverse. However, this process is not shown in the diagram. Lastly, Fig. 1 also presents the architecture of HIDS as labeled on individual devices, which administers further packet inspection within the host (*Vokorokos & Baláž, 2010*), thereby enhancing the security level of a given network or computer system.

## Summary of Snort and its components

Snort is a popular and influential cross-platform lightweight signature-based network intrusion detection and prevention system with multiple packet tools. The power of Snort

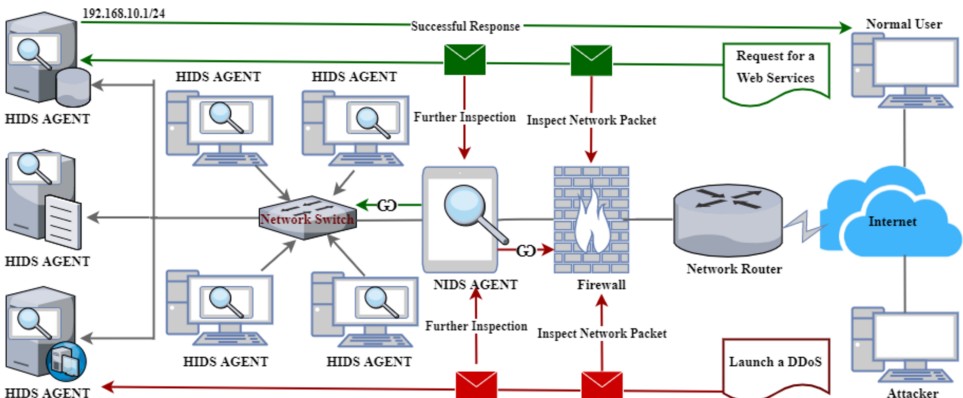

**Figure 1** **A classical architecture of host and network-based intrusion detection system.**

lies in the use of rules, some of which are preloaded, but we can also design customized rules to merely send alerts or block specific network traffic when they meet the specified criteria. Additionally, alerts can be sent to a console or displayed on a graphical user interface, but they can also be logged to a file for future or further analysis. Finally, Snort also enables the configuration options of logging alerts to databases such as MSQL and MongoDB or sending an email to a specified responsible person if there are alerts or suspicious attempts (*Ali et al., 2018*).

Moreover, Snort has three running modes: sniffer, packet logger, and network-based IDS mode (*Tasneem, Kumar & Sharma, 2018*). The sniffer mode is run from the command line mode, and its primary function is just inspecting the header details of packets and printing it on the console. For instance, *./Snort –vd* will instruct Snort to display packet data with its headers. The packer logger mode interestingly inspects packets and logs them into a file in the root directory. Then, it can be viewed using tcpdump, snort, or other applications for further analysis. For example, *./Snort -dev -l ./directory_name*, will prompt Snort to go to packet logger mode and log the packets in a given directory; if the *directory_name* did not exist, then it will exit and throw an error. Finally, the network-based IDS mode utilizes the embedded rules to determine any potential intrusive activities within a given network. Snort does this with the help of the network interface card (NIC) running in the promiscuous mode to intercept and analyze real-time network traffic. For instance, the command *./Snort -dev -l ./log -h* 172.162.1.0/24 − *c snort.conf* will prompt Snort to log network packets that trigger the specified rules in *snort.conf*, where *snort.conf* is the configuration file that applies all the rules to the incoming packets for any malicious attempt (*Tasneem, Kumar & Sharma, 2018*).

It is equally important to note that Snort also has the strength of real-time packet logging, content matching, and searching with protocol analysis. It also has the advantage of serving as a prevention tool instead of just monitoring (*Ali et al., 2018*). However, it does come with some notable shortcomings. For example, a very unpopular GUI makes using Snort a bit difficult. Additionally, the vast number of network traffics can compromise the reliable and functional operation of Snort, which inspires the use of Pfring and Hyperscan

to reinforce the functionality of Snort for efficiency and reliability. Finally, and most importantly, caution needs to be exercised in creating Snort rules to avoid the apparent challenge of many false alarm rates (FAR) (*Park & Ahn, 2017*). Snort comprises five logical components that determine and classify potential malicious attacks or any undesired threat against computer systems and networks (*Ali et al., 2018*). Finally, interested readers can refer to the following references for the details of the Snort components (*Essid, Jemili & Korbaa, 2021*; *Mishra, Vijay & Tazi, 2016*; *Shah & Issac, 2018*).

## Snort rule syntax

Snort utilizes a flexible, lightweight, and straightforward authoritative rules-language primarily written in a single line as in versions preceding to 1.8. However, the present Snort versions allow the spanning of rules in multiple lines but require the addition of backslash (\) at the end of each line; generally, Snort rules are composed of dual logical portions. For instance, the rule header and the rule options (*Khurat & Sawangphol, 2019*). Usually, Snort rules share all sections of the rule option like the general options, payload detection options, non-payload detection options, and post-detection options. However, it can be specified differently depending on the configuration approach. Finally, Snort rules are generally applied to the headers of the application, transport, and network layers such as FTP, HTTP, ICMP, IP, UDP, and TCP (*Chanthakoummane et al., 2016*; *Khurat & Sawangphol, 2019*). However, they can also be applied to the packet payload, which is the adopted approach for SARG.

Table 1 presents a classical representation of Snort rules (*Khurat & Sawangphol, 2019*). The two rules shown in Table 1 denote that an alert will be triggered based on an icmp traffic protocol from *any* source IP address and port number to *any* destination IP address and port number if the traffic content contains a *probe*. Consequently, this will show a message *probe attack*, and the signature ID of this rule is 1000023. Similarly, the second rule is almost the same as the first, except that the action is "*log*" instead of "*alert*", while the destination port number is 80 instead of "*any*" number.

## The rule header

The Snort rule header comprises the specified actions, protocol, addresses of source and destination, and port numbers. The default Snort actions are: *alert*, *log*, and *pass*, and it is a required field for every Snort rule, and it defaults to *alert* if not specified explicitly. Nevertheless, *drop*, *reject*, and *sdrop* are additional options for an inline mode (*Chanthakoummane et al., 2016*; *Khurat & Sawangphol, 2019*).

The protocol field within the rule header is required and usually defaults to IP. Nonetheless, it equally supports UDP, TCP, and ICMP options. The source IP is an optional field and, by default, is set to *any*, as indicated in Table 1. However, it also supports a single IP address like 172.168.10.102 or a CIDR block like 172.168.10.0/16, which permits a range of IP addresses as an input. Likewise, a source port field allows a port number or range of port numbers, and the destination IP and port fields are almost the same as the source IP and port fields. Finally, the direction of the monitored traffic is specified using the directional operators (->, and <-), and the monitoring of source to destination (->) is the most common practice (*Khurat & Sawangphol, 2019*).

**Table 1  A classical representation of Snort rules syntax.**

| Action | Protocol | Src. IP | Src. Port | Direction | Dst. IP | Dst. Port | Parameters | | |
|--------|----------|---------|-----------|-----------|---------|-----------|------------|---|---|
| alert | tcp | icmp | any | -> | any | any | (msg: "probe attack"; | content: "probe"; | sid: 1000023) |
| log | tcp | any | any | -> | any | 80 | (msg: "HTTP attack"; | content: !"GET"; | sid: 1000024) |

**Table 2  A typical representation of the rule header components.**

| Action | Protocol | Source. IP | Source port | Direction | Destination IP | Destination port |
|--------|----------|------------|-------------|-----------|----------------|------------------|
| alert | tcp | any | :1024 | -> | 192.168.100.1 | 600: |
| log | icmp | !172.16.10.0/16 | any | -> | 172.16.10.252 | 1:6000 |
| alert | tcp | !192.168.10.0/24 | any | <> | 192.168.10.0/24 | 1:1024 |

Table 2 presents the rule header components with typical examples. For example, the first example will trigger an alert for traffic from *any* source address with various port numbers up to and including 1024 denoted (:1024), which is going to 192.168.100.1, and ports that are greater than and including port 600 denoted (600:). Finally, the second and last examples of Table 2 are almost the same as the first except for the change in specific fields.

## The rule options

The rule option unit comprises the detection engine's central functionalities yet provides complete ease of use with various strengths and flexibility (*Khurat & Sawangphol, 2019*). However, this segment can only be processed if all the preceding details have been matched. Furthermore, since this unit generally requires a vast amount of processing resources and time, it is recommended to limit the scope of the rules using only the necessary fields to enable real-time processing without packet drops. Therefore it is recommended to only use the *message*, *content,* and *SID* field for writing efficient and reliable rules. The semicolon (;) separates the rule options while the option keywords are separated using the colon (:). Finally, this unit comprises four main classes, but this paper will only summarize the general and payload detection options. For instance, the general options have no significant effect during detection but merely provide statistics about the rule, whereas the payload inspects the packet data. The general rule and payload detection options include numerous parameters available to Snort users for rule creation (*Khurat & Sawangphol, 2019*). However, only the relevant ones are selected to understand the work proposed in this paper, and Table 3 presents the selected options with a typical example for each.

## Snort configurations and rule files

Snort provides a rich scope of customizable configuration options for effective deployment and day-to-day operations. Since Snort consists of vast configuration options, this section will only highlight the necessary options to understand this work easily. Generally, the *snort.conf* contains all the Snort configurations, and it includes the various customizable

**Table 3** A typical illustration of the rule header components.

| General rule options | | | | | Payload options | | |
|---|---|---|---|---|---|---|---|
| msg | sid | rev | gid | priority | content | offset | depth |
| (msg:"Anomaly detected"; | sid: 1000027; | rev: 3; | gid: 1000032; | priority: 1; | content: "probe-attack"; | offset:4; | depth:12; |

settings and additional custom-made rules. The *snort.conf* is a sample and default configuration file shipped with the Snort distribution. However, users can use the -c command line switch to specify any name for the configuration file, such as */opt/snort/Snort -c /opt/snort/myconfiguration.conf*. Nevertheless, *snort.conf* is the conventional name adopted by many users (*Erlacher & Dressler, 2020*). Likewise, the configuration file can also be saved in the home directory as *.snortrc* but using the configuration file name as a command-line is the common practice with advantages. Moreover, it also enables using variables for convenience during rule writing, such as defining a variable for HOME_NET within the configuration file like *var HOME_NET 192.168.10.0/24*. The preceding example enables the use of HOME_NET within various rules, and when a change is needed, the rule writer only needs to change the variable's value instead of changing all the written rules. For instance, *var HOME_NET [192.168.1.0/24,192.168.32.64/26]* and *var EXTERNAL_NET any* (*Mishra, Vijay & Tazi, 2016*).

Lastly, the rules configuration also enables the Snort users to create numerous customized rules using the variables within the configuration files and add them to the *snort.conf* file. The general convention is to have different Snort rules in a text file and include them within the *snort.conf* using the *include* keyword like *include $RULE_PATH/myrules.rules*, which permits the inclusion of the rules within *myrules.rules* to the *snort.conf* file during the next start of snort. Also, users can use the Snort commenting syntax (#) in front of a specified rule or the rule file within the Snort configuration file to manually disable the Snort rule or the entire class of rules.

Similarly, Snort parsed all the newly added rules during a Snort startup to activate any newly added rules. However, if there are any errors within the newly added rules, Snort will exit with an error, necessitating the correct and consistent writing of Snort rules (*Erlacher & Dressler, 2020*; *Mishra, Vijay & Tazi, 2016*). Finally, Figs. S1 and S2 are the standard representations of the Snort configuration and rule files.

## Synopsis of alert correlation

Alert Correlation is a systematic multi-component process that effectively analyzes alerts from various intrusion detection systems. Its sole objective is to provide a more concise and high-level view of a given computer system or network, and it can prioritize IDSs alerts based on the severity of the attack. Additionally, it plays a significant role for the network administrators to effectively and efficiently differentiate between relevant and irrelevant attacks within a given network, resulting in reliable and secured networks (*Valeur, Vigna & Kruegel, 2017*).

# MATERIALS AND METHODS

The section briefly discusses the methodologies and procedures used to design and implement the proposed Snort Automatic Rule Generator using a sequential pattern algorithm and the Security Event Correlation, abbreviated SARG-SEC. It is worthy to note that the term event is the same as alerts generated during intrusion detection.

## Snort automatic rule generation (SARG)

Irrespective of the significant numbers of available literature that use other approaches of automating rule generation with remarkable success (*Li et al., 2006*; *Ojugo et al., 2012*; *Sagala, 2015*), there is still a need for an effective and optimized auto-rule generator. According to the authors' knowledge, none of the existing literature uses the proposed approach in this paper. Therefore, this paper presents an automation method of generating content-based Snort rules from collected traffic to fill this research gap. The following sections provide a concise description of the design and implementation procedures.

## Content rule extraction algorithm

Snort rules can indicate various components, but this research targets rules that only include header and payload information. The steps demonstrated in the examples below are performed to generate the Snort rules automatically. For each first host, the application and service to be analyzed or the traffic of malicious code is collected. Moreover, packets with the same transmission direction are combined to form a single sequence in a single flow. Finally, the sequence is used as an input of the sequential pattern algorithm to extract the content (*Pham et al., 2018*).

*Action Protocol SourceIP SourceProt—>DestinationIP DestinationPort (Payload)*

*Alert tcp 192.168.0.0 192.168.0.0/24—>IP 80 (content: "cgi-bin/phf"; offset:6; depth:24)*

The sequential pattern algorithm finds the candidate content while increasing the length starting from the candidate content whose length is 1 in the input sequence and finally extracts the content having a certain level of support. Nevertheless, the exclusive use of only packet or traffic contents for rule creation can lead to significant false positives, thereby undermining the effectiveness of the created rules and increasing the potential risks against computer systems and networks. Therefore, additional information is analyzed and described in the rule to enable efficient and effective rules against malicious attacks. As a result, this paper used the content and header information of the network traffic or packets as supplementary information, thereby significantly enhancing the reliability of the automatically generated rules at the end of the process. Moreover, by applying the extracted content to input traffic, traffic matching the content is grouped, and the group's standard header and location information is analyzed (*Sagala, 2015*). Finally, the auto-generated Snort rules are applied to the network equipment with Snort engine capability.

## Sequence configuration Steps

The sequence is constructed by only extracting the payloads of the packets divided into forward and backward directions of the flow. Moreover, the proposed approach will generate two sequences for flows that consist of two-way communication packets, whereas

the unidirectional communication traffic generates a single sequence. Finally, it is worthy to note that a sequence set consists of several sequences denoted **S** as shown in Eq. (1), and a single sequence is made up of host ID and string as shown in Eq. (2).

$$\text{Sequence Set} = \{S_1, S_2, \ldots., S_s\} \tag{1}$$

$$S_i = \{host\_id, < a_1 a_2 a_3, \ldots., a_n >\}. \tag{2}$$

## Content extraction step

A sequence set and a minimum support map are input in the content extraction step and extract content that satisfies the minimum support. This algorithm improves the Apriori algorithm, which finds sequential patterns in an extensive database to suit the content extraction environment. The Content Set, which is the production of the algorithm, contains several contents (C) as shown in Eq. (3), and one content is a contiguous substring of a sequence string as shown in Eq. (4). Algorithm 1 and Algorithm 2 illustrate producing a content set that satisfies a predefined minimum map from an input sequence set. Moreover, when Algorithm 1 performed the content extraction, the content of length one (1) is extracted from all sequences of the input sequence set and stored in a content set of length 1 ($L_1$) as demonstrated in (Algo.1 Line: 1~5), the content of length 1 starting with the length is increased by 1. Also, the content of all lengths is extracted and stored in its length content set ($L_k$) as shown in (Algo.1 Line: 6~20).

$$\text{Content Set} = \{C_1, C_2, \ldots, C_c\} \tag{3}$$

$$C_i = \{< a_x a_x + 1 \ldots. a_y > | 1 \leq \times \leq y \leq n, \} \tag{4}$$

$$\text{Support} = \text{Number of support hosts / total number of hosts.} \tag{5}$$

It is worthy to note that the method to be used at C (Algo. 1.0 Line: 18) is the process described in Algorithm 2.0.

**Algorithm 1.0:** Content extractor (CE) based algorithm

**Input:** *SequenceSet = {$S_1$, $S_2$... $S_s$}, minimumsupport*

**Output:** ContentSet = {$C_1$, $C_2$, …. $C_c$}

CE (SeqSet, MinimumSupport)

```
01 begin
02      for each Seq S in the Seq Set do;
03              for each Character a in the S do
04                      L₁ = L₁ Uₐ;
05              END // blocks of statements
06      END // blocks of statements
06      Kₒ=2.0;
07      while  L_{k-1} = O do
08              for each content L_{k-1}  do;
09                      for I=1  to Sequence do;
10                              if (Sᵢ include c);
11                                      Count = Count +1;
12                              END// blocks of statements
13                      END// blocks of statements
14                              if ((count /s) < minsupp)
15                                      L_{k-1} = L_{k-1} –C;
16                              END// blocks of statements
17                      END// blocks of statements
18                      L_k=candi_gene (L_{k-1})
19                      K₀++;
20      END// blocks of statements
21      ContentSet = A_{lk}
22       Return Content Set
23 end;
```

***Representation**: $L_k$: Length k ,CE: Content extractor*

**Algorithm 2.0:** The sub content extract (SCE) based algorithm

**Input:** $L_{k-1}$

**Output:** $L_k$

*Candi_gen(Lk-1),*

```
01 begin
02      for each content p0 in L_{k-1} do;
03              foreach content q₀ in L_{k-1} do
04                      if ((p.a₂ = q.a₁) && (p.a₃ = q.a₂) && (p.a_{k-1} = q.a_{k-2}) ) then;
05                              l_k – l_k U < p.a₁, p.a₂,     , p.a_{k-1},q.a_{k-1} >;
06                      END // blocks of statements
06              END // blocks of statements
07      END // blocks of statements
08      Return L_k
09 end;
```

***Representation:** $L_k$ : Length K , CE: Content Extractor*

The contents of the set $L_{k-1}$ are created by comparing the contents of $L_k$. Furthermore, to make the content of the set $L_{k-1}$ by combining the contents of the set $L_k$, contents of the set with the same length *k-2* content excluding first character and *k-2* content excluding

the last character are possible as shown at (Algo.2.0 Line: 1~7). For example, *abcd* and *bcde* are the contents of the set *L4*. It has the same *bcd* except for *a,* and *bcd* except for *e*. Therefore, the content *abcde* of the set L5 can be created by increasing the length by 1 in the same way as above. Extracting and deleting content below the support level are repeated until the desired outcome.

The final step checked for the content inclusion relationship of all lengths extracted. If we find the content in the inclusion relationship, the content is deleted from the set as demonstrated in (Algo.1.0). Then, we deliver the final created content set to the next step. Moreover, the SequenceSet consisting of traffic from 3 hosts and minimum support of 0.6 is passed as an input. The minimum support rating of 0.6 means that since the total number of hosts is 3, the content must be observed in traffic generated by at least two hosts. Finally, all length one (1) content is extracted when the algorithm is executed.

Furthermore, Algorithm 3.0 would have been given content and packet set when it represents analyzing the location information of the content. The output of this algorithm is offset when matched to a packet in a packet set. The matching starts in a bit, byte position, and depth is the matching exit position, meaning the maximum byte position.

| **Algorithm 3.0:** Locate data extract (LDE) based algorithm |
|---|
| **Input:** *Content, Packet set = { p₁, p₂, pₙ }* |
| **Output:** *Address location, Depth* |
| *ACL (cont , packet set),* |
| 01 **begin** |
| 02      *AL = Maximum_ Packet_ Size;* |
| 03      *Dept =0;* |
| 04      *for each packet t in packet Set do;* |
| 05          *if(t.isMatchContent(cont)) then* |
| 06            *al = minimum (offset, t.getStartMatch(cont));* |
| 07            *depth=maximum (offset, t.getStartMatch(cont));* |
| 06          *END // blocks of statements* |
| 07      *END // blocks of statements* |
| 08      *Return address location, depth;* |
| 09 **end**; |
| ***Representation:** ACL = analysis content location Al= depth maximum byte point each match* |

The first offset is the maximum size of the packet, and the depth initializes to 0 as shown in (Algorithm 3.0 Line: 1~7), which traverses all packets in packet set and adjusts offset and depth. Moreover, it checks whether the content received matches the packet, and if there exists a match, then the starting byte position is obtained and compared to the current offset. If the value is less than the current offset, the value changes to the current offset. Similarly, in the case of depth, if the value is more significant than the current dept., the current value is obtained using the byte position. Then it changes the value to the current depth (Algo.3 Line: 4~6). Likewise, to analyze the header information of the extracted content, we performed a process similar to the location information.

The analysis steps described above traversed all packets in the packet set and checked for possible matching with the content. If a match exists, it will store the packet's header information. After reviewing all packets, it adds the header information to the content rule if the stored header information has one unique value. Additionally, in IP addresses, the CIDR value is reduced to 32, 24, 16 orders which iterate until the unique value is extracted. Assuming the CIDR value is set to 32, which is the class D IP address range, we will try to find an exclusive value, and if not found, then apply the CIDR value to 24 to find the class C IP address, and this process continues until we obtained the desired values. For instance, if the Destination IP address to which that content is matched is CIDR 32, then "111.222.333.1/32" and "111.222.333.2/32" are extracted. In contrast, it can also be set to CIDR 24 and extract "111.222.333.0/24".

### The schematic diagram of the snort automatic rule generator (SARG)

Figure 2 presents a comprehensive schematic diagram of SARG. The proposed auto-rule generator (SARG) utilized well-known datasets used by many researchers and security experts to assess various network security frameworks as discussed in *Lippmann et al. (2000)*. Initially, we used different pcap files to simulate live attacks against SARG that facilitates an efficient and effortless generation of reliable Snort rules. It is essential to note that $\pi_{-1}, \pi_{-2}, \pi_{-3}, \ldots, \pi_{-n}$, denotes the various utilized pcap files for the auto-rule generation. Consequently, SARG relies on the pcap file contents to automatically generate numerous effective Snort rules without any human intervention, and $\lambda_{-1}, \lambda_{-2}, \lambda_{-3}, \lambda_{-4}, \ldots, \lambda_{-n}$ represents the various auto-generated Snort rules. Next, the Snort.conf file is updated based on the auto-generated rules. In addition, any device with a Snort engine denoted as $\bar{\omega}$ can use these auto-generated rules against incoming traffic represented as $D_{-1}, \ldots, D_{-n}$ to trigger alerts for malicious attempts that meet all the criteria of the rules. Finally, this paper does not document the generated alerts due to the volume of the work. However, the provided supplementary materials contain the codes and pcap files that any interested researcher can reproduce.

### Overview and significance of alert correlation

Intrusion detection systems have recently witnessed tremendous interest from researchers due to their inherent ability to detect malicious attacks in real-time (*Vaiyapuri & Binbusayyis, 2020*; *Zhou et al., 2020*). In addition, it has significantly mitigated the security challenges that came with ground-breaking technologies such as the Internet of Things (IoT) (*Verma & Ranga, 2020*), big data, and artificial intelligence (*Topol, 2019*). However, regardless of the immense contributions mentioned above, it may still generate thousands of irrelevant alerts daily, which complicates the role of the security administrators in distinguishing between the essential and nonessential alerts such as false positives.

In addressing these security challenges, specifically for IDSs, the research industry has proposed significant pieces of literature with remarkable achievements in lessening the huge false alarm rates of current cutting-edge proposed systems (*Mahfouz et al., 2020*; *Zhou et al., 2020*). Nevertheless, irrespective of these significant achievements, there is still a need to propose new approaches with much efficiency and effectiveness to help mitigate

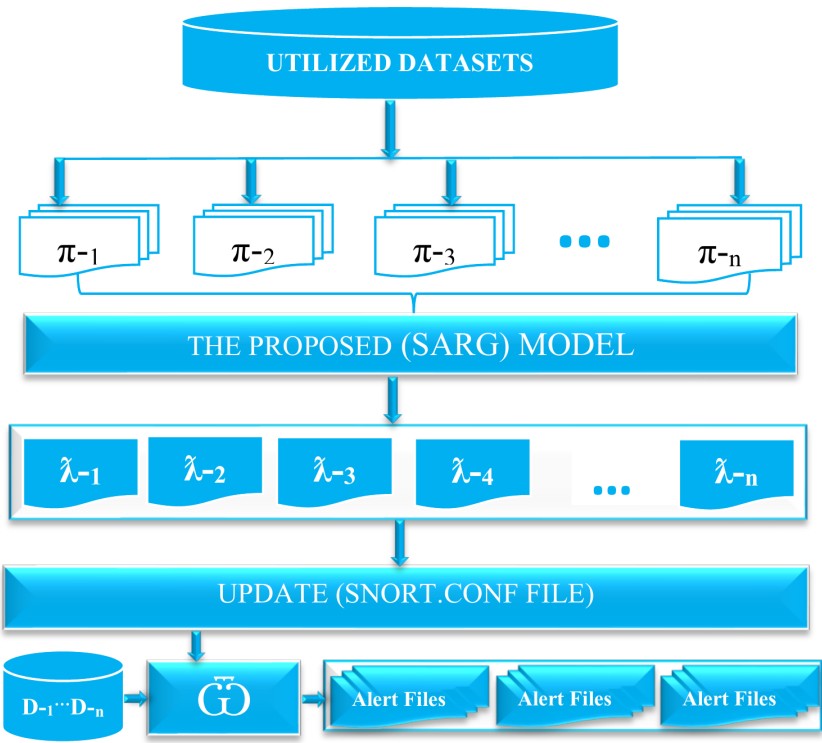

**Figure 2** **The schematic diagram of the snort automatic rule generator (SARG).**

the vast false alarm rates of the earlier proposed systems (*Jaw & Wang, 2021*). Therefore, this paper presents a novel alert correlation model that correlates and prioritizes IDS alerts. The following sections provide a succinct description of the design and implementation process of the proposed security event correlation model.

## The proposed security events correlation (SEC)

The SEC model offers efficiency and effectiveness of correlating the alerts generated by the IDS, which significantly mitigates the massive false alarm rates. Furthermore, it does not need previous knowledge while comparing different alerts to measure the similarity in various attacks. Unlike other correlation approaches that usually follow specific standards and procedures (*Valeur, Vigna & Kruegel, 2017*; *Zhang et al., 2019*), we proposed a model that entails the same processes but with different approaches, as shown in the overview of the proposed model below.

The overview of the proposed system involves various phases such as monitoring interval, alert preprocessing, alert clustering, correlation, alert prioritization, and the results as shown in Fig. 3. The following sections detail the various phases of SEC with their respective procedures.

## The monitoring interval

In summary, this paper defined the monitoring interval as the specific configured time on the security framework or monitoring model performing the alert correlation. The

**Figure 3  An overview of the proposed security event correlator (SEC).**

monitoring interval of SEC is 300 days. It is essential to note that the long interval of 300 days is because more days means extra alerts to correlate and aggregate, leading to effective and better results.

## Alert preprocessing

The alert preprocessing stage of this work involves two sub-processes, namely, feature extraction and selection. This phase systematically extracts features and their equivalent values from the observed alerts. The similarity index defined below calculates the alerts with the same value for existing features such as attack category, detection time, source IP address, and port number. Consequently, it significantly helps identify new features from the available alerts, thereby overcoming attack duplication. Also, this phase involves the selection of features for alert correlation. For example, suppose the selected features such as the attack detection time, category, port number, and source IP address of one alert or multiple alerts are the same. In that case, we use the similarity index to select the relevant alerts and eliminate irrelevant ones such as duplications and similar instances. Thus, $\Pi$ denotes the similarity between two alerts, and $\omega$ represents the alert similarity index, and $\delta$ denotes an alert.

$$\Pi = \omega(\delta a, \delta b), where \delta a \neq \delta b. \tag{6}$$

Lastly, this section also involves alert scrubbing that uses attack type, detection time, source IP address, and detection port to remove the incomplete alert data. This process plays a significant role in providing reliable and consistent data.

## Alert aggregation and clustering

The existing literature has provided various descriptions of alert aggregation, such as considering alerts to be similar if all their attributes match but with a bit of time difference. In contrast, others extended the concept to grouping all alerts with the exact root causes by aggregating alerts using various attributes. Moreover, this phase groups similar alerts based on the similarity index of the extracted features. For instance, if the features of alert $\bar{Y}$ have the same value as alert $\ddot{X}$, then this phase will automatically convert these two alerts into a single unified alert by removing the duplicated alerts, thereby significantly reducing the number of irrelevant alerts. Finally, the clustered alerts provide an effective and efficient analysis of false positives, leading to reliable and optimized security frameworks.

Lastly, Fig. 4 demonstrates the alert aggregation process. Firstly, we cluster all the generated alerts, check their similarity using the similarity index defined in previous sections, and remove all the duplicated alerts.

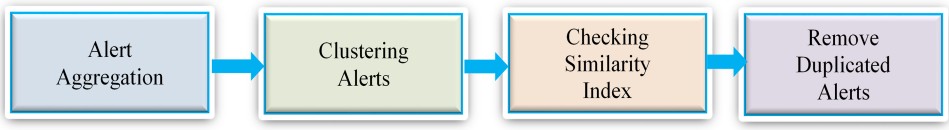

**Figure 4** **Illustration of the alert aggregation process.**

## Correlation

Based on existing literature, the primary objective of alert correlation is to identify underlying connections between alerts to enable the reconstruction of attacks or minimize having massive irrelevant alerts. Furthermore, scenario-based, temporal, statistical, and rule-based correlations are among the most commonly utilized correlation categories in existing research. The approach in this paper has fully met the attributes of the statistical correlation method, which correlates alerts based on their statistical similarity. Finally, this phase consists of the alert correlation process based on selected features like the IP addresses, attack category, and the detected time of the generated alerts. For example, if the detection time of two independent alerts satisfies the condition of COTIME denoted as $\omega$, then the correlation is performed based on selected features.

Figure 5 presents the correlation engine with the extracted features for alert correlation. The correlation engine used these features to efficiently correlate alerts that meet the conditions of the $\omega$.

## Alert prioritization

Alert prioritization is the final phase of the proposed SEC model. It plays a significant role in prioritizing the alert's severity, thereby helping the administrator identify or dedicate existing resources to the most alarming malicious attacks. Although a priority tag is assigned to each Snort rule, as explained earlier, we intend to extend its functionality by embedding an alert prioritization within the proposed SEC model. Alert prioritization is classified into high, medium, and low priority. Firstly, this paper defines the high priority as the alert counts with standard features such as the data fragmentations, source IP address, port number, and attack category.

For instance, if multiple alerts have the same destination port numbers and IP addresses, then we classify these alerts as high-priority alerts. Lastly, the high priority functionality will hugely minimize various challenging cyber-attacks such as DDoS and DoS because it handles the standard techniques utilized in these attacks, like the IP fragmentation attack, which uses the analogy of data fragmentation to attack target systems. Secondly, the medium priority alerts count the number of alerts with shared features such as the same attack category, IP addresses, and destination port number. Finally, low priority alerts are alert counts with standard features like IP addresses, attack category, and destination port but varying values.

The alert prioritization categories with the various standard features discussed above are shown in Fig. 6. However, it is essential to note that irrespective of the common features

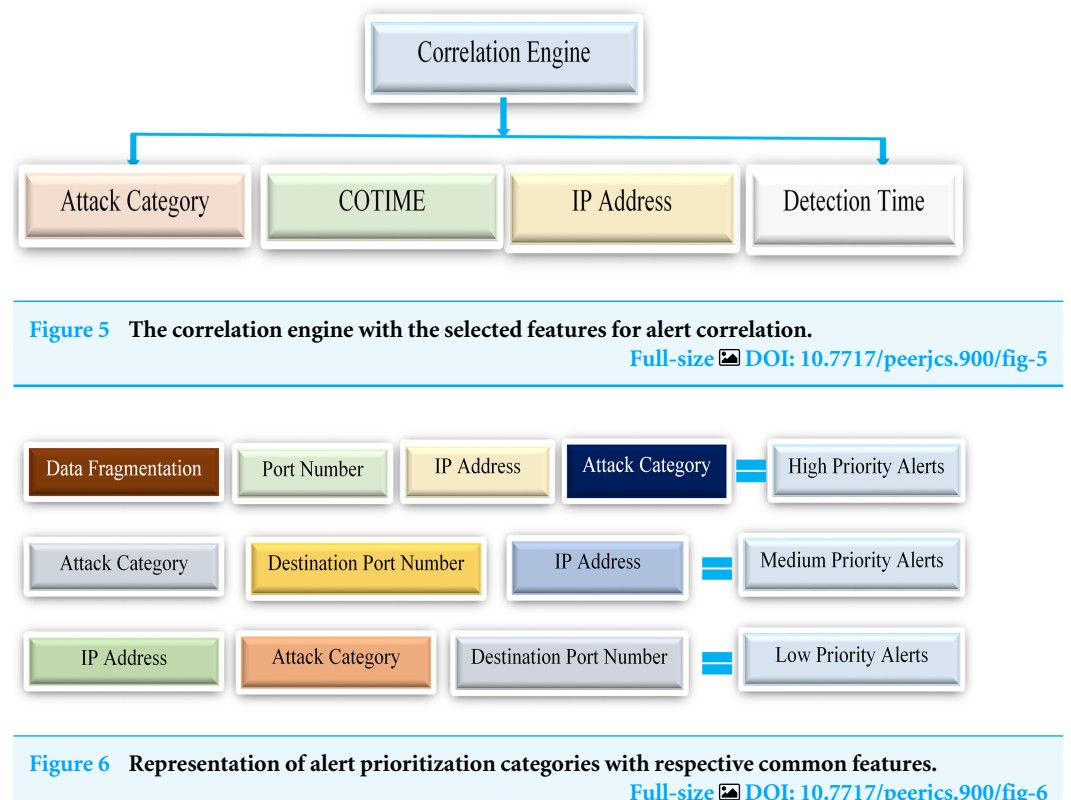

**Figure 5** The correlation engine with the selected features for alert correlation.

**Figure 6** Representation of alert prioritization categories with respective common features.

in each category, the above section entails specified conditions that uniquely distinguish them.

## Overview of the proposed security events correlation (SEC)

Figure 7 demonstrates the detailed overview of the proposed SEC. First of all, SEC accepts a collection of raw alerts as an input generated based on a specified monitoring interval set to 300 days to ensure sufficient alerts for practical analysis. Additionally, the alert preprocessing step accepts these raw alerts and performs feature extraction and selection. Furthermore, the alert scrubbing phase uses predefined conditions denoted as $\Xi$, to check for alert duplicates. If duplicates exist, the scrubbing process will remove all the copies and send the alerts with no duplicates to the correlation engine. Next, the correlation engine decides if the alerts are single or multiple instances using predefined conditions denoted as $<= \omega$. Finally, the alert prioritization phase prioritizes the alerts using the above three categories, and the output phase presents the relevant alerts as the final results. It is worthy to note that the alert outputs are the actual events after removing the duplicates and the alert correlation, as shown in the equation below.

$$\Omega = \pi - \beta \tag{7}$$

Where $\Omega$ denotes the final output alerts, $\pi$ represents the total number of alerts, and $\beta$ denotes the correlated alerts.

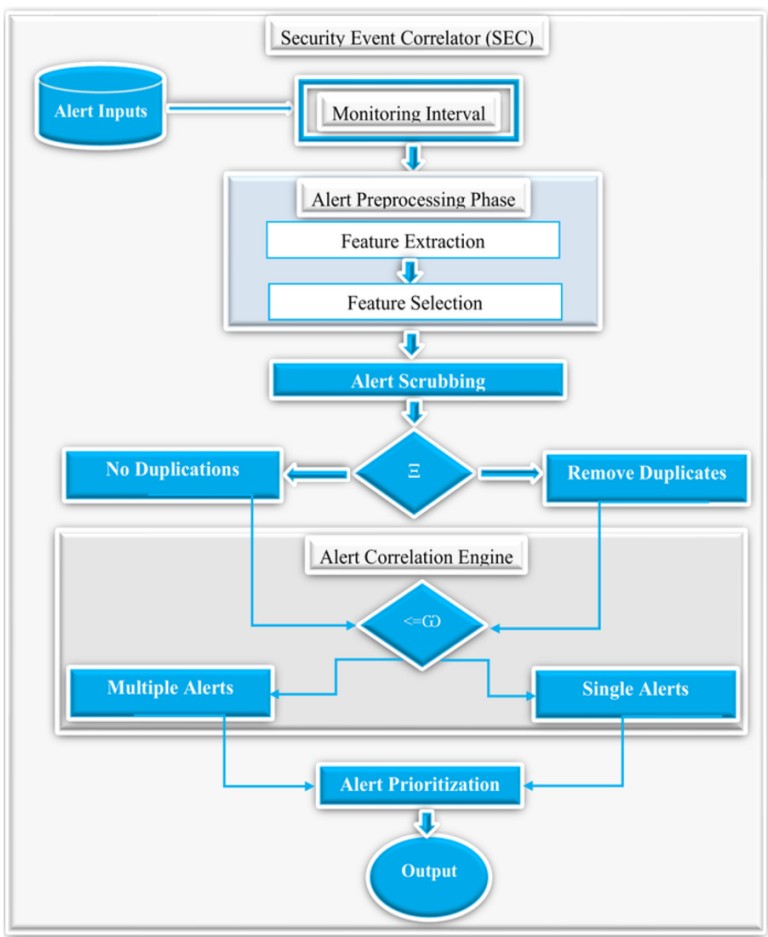

**Figure 7** The detailed overview of the proposed security event correlator (SEC).

## RESULTS AND DISCUSSION

This section presents a comprehensive systematic analysis and performance justification of the proposed models (SARG-SEC). For example, it evaluates how well the SARG can efficiently generate standard and reliable Snort rules by executing SARG against live attacks in existing pcap files. Finally, this section will also highlight the results and performance evaluation of the (SEC) model that significantly mitigates the challenges of the vast alerts generated by the Snort IDS and the earlier proposed feature selection and ensemble-based IDS (*Jaw & Wang, 2021*).

### Evaluation of the generated Snort rules by the proposed SARG

The authors of *Lippmann et al. (2000)* provided a descriptive and intriguing challenge of documenting an off-line IDS dataset that a wide range of security experts and researchers heavily utilized to assess various security frameworks. Moreover, the paper presents a testbed that used a tcpdump sniffer to produce pcap files used to evaluate the proposed solutions (SARG-SEC). Finally, the supplementary pcap files also entail the various

**Figure 8 Auto generated snort rules using pcap files as live attacks.**

attacks utilized to evaluate SARG-SEC. Accordingly, this section highlights the findings of the various conducted experiments to accurately assess the performance of the SARG framework, as demonstrated in Figs. 8 & 9.

The proposed SARG method has achieved decent performances on the auto-generation of Snort rules, as illustrated in Fig. 8. All the findings presented in this section use various pcap files as a simulation of live attacks against the proposed method to generate efficient and effective Snort rules that completely meet all the criteria of the Snort rule syntax, as presented in Table 3. For instance, Fig. 8 demonstrated that SARG has successfully auto-generated a Snort rule with the following content: *alert udp 10.12.19.101 49680 ->any any (msg: "Suspicious IP10.12.19.101 and port 49680 detected"; reference:Packet2Snort; classtype:trojan-activity, sid:xxxx, rev:1)*. Based on the above auto-generated Snort rule, it is self-evident that SARG has produced optimized Snort rules that meet all the criteria of Snort rule syntax as discussed in *Khurat & Sawangphol (2019)*. For example, the above auto-generated rule has produced a descriptive and easy to analyze message of (*msg: "Suspicious IP10.12.19.101 and port 49680 detected ";),* which details the reason or cause of the alert.

Furthermore, SARG auto-generated another similar Snort rule by setting the source IP address and port number to "*any any*" and the destination IP address and port number as *10.12.19.1* and *53*, respectively. Lastly, the findings presented in Fig. 8 shows that SARG has auto-generated an impressive and comprehensive Snort rule that meets all standard Snort rule criteria. However, the auto-generated alert uses the *HOME_NET* variable as the source address and a *msg, content, dept*, and *offset* values of (*msg: "Suspicious DNS request for fersite24.xyz. detected"; content:" |01000001000000000000|"; depth:10; offset:2;),* respectively. Based on the above performances, the authors concluded that SARG has effectively met all the criteria of creating compelling and optimized Snort rules consisting of numerous general rule parameters and payload detection options (*Khurat & Sawangphol, 2019*).

Moreover, Fig. 9 presents a series of auto-generated Snort rules using several pcap files as live attacks. All the auto-generated Snort rules shown in Fig. 9 have completely demonstrated to meet all the standards of Snort rule creation, as discussed in much existing work (*Jeong et al., 2020*; *Khurat & Sawangphol, 2019*).

**Figure 9** A demonstration of auto-generated snort rules using various pcap files as live attacks.

For instance, the use of defined variables like the *HOME_NET* and ASCI values of (*"|05|_ldap|04|_tcp|02|dc|06|_msdcs|0C|moondustries|03";)* for the values of *"content:"* field of the auto-generated Snort rules. Also, all the automatically generated rules came with auto-generated messages that uniquely and vividly describe the intrusive or abnormal activity whenever the alert is triggered. Consequently, this will significantly help the administrator to be able to easily identify why the alert happened and quickly find solutions to mitigate any malicious activities.

Additionally, all the auto-generated Snort rules are saved in the *local.rules* to ensure consistency and avoid duplications of rules within the *local.rules* files, and then included in the *snort.conf* file. As a result, it provides the administrator with the flexibility to further fine-tune the generated rules to meet the specific needs of a given computer network or system. Thus, irrespective of the additional efforts of fine-tuning the auto-generated rules, for example, manually updating the *sid* value of the auto-generated rules and other fields for specificity, SARG has undoubtedly minimized the time taken for Snort rule creation. Also, it can significantly lessen the financial burden of human experts for Snort rule creation, thereby making the proposed method a meaningful tool that would play a significant role in mitigating the escalating cyberattacks.

In conclusion, the results presented in Figs. 8 and 9 have impressively demonstrated the capability of auto-generating Snort rules, which considerably mitigates the need for

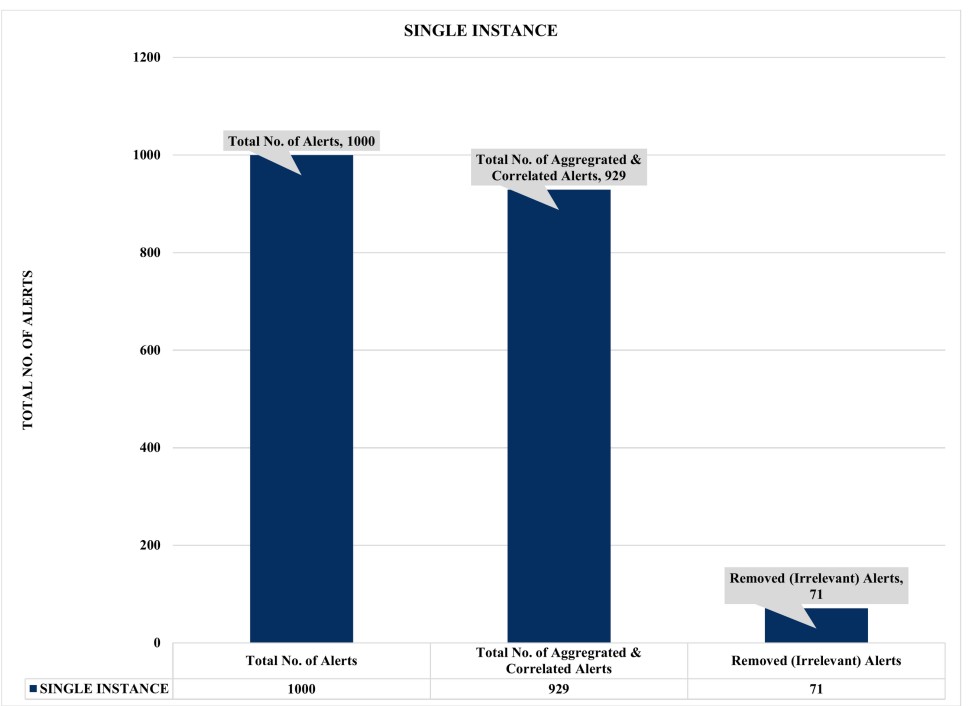

**Figure 10** Evaluation of single instance alert aggregation and correlation.

costly human capacity in creating Snort rules. However, it has downsides that we could not handle, such as automatically generating the consistent *sid* values of the respective rules. As a result, we came up with a prompt message to remind the users to update the *sid* value after the auto-generation of the Snort rule. Also, based on the lack of expert domain, we could not justify why SARG has two contents in a single generated Snort rule. Nevertheless, these challenges have no negative implications while validating the auto-generated rules. Regardless, we intend to extend this research to establish means of solving these research challenges.

## Analysis of the single and multiple instances security event correlations

Likewise, the following sections present the results of numerous experiments that evaluate the consistency and efficiency of the proposed security event correlator (SEC). Also, the findings presented in the subsequent sections have demonstrated the promising performances of the SEC model, which could significantly mitigate the substantial challenges of managing the vast alerts generated by heterogeneous IDSs.

Firstly, Figs. 10 and 11 summarized the results of the experiments based on single and multiple instances. For example, Fig. 10 depicts a single sample of one thousand (1000) alerts used to evaluate the effectiveness of the proposed security event correlator (SEC), which obtained an impressive correlation performance of removing 71 irrelevant alerts due to alert duplication. Even though the 71 removed irrelevant alerts might seem insignificant,

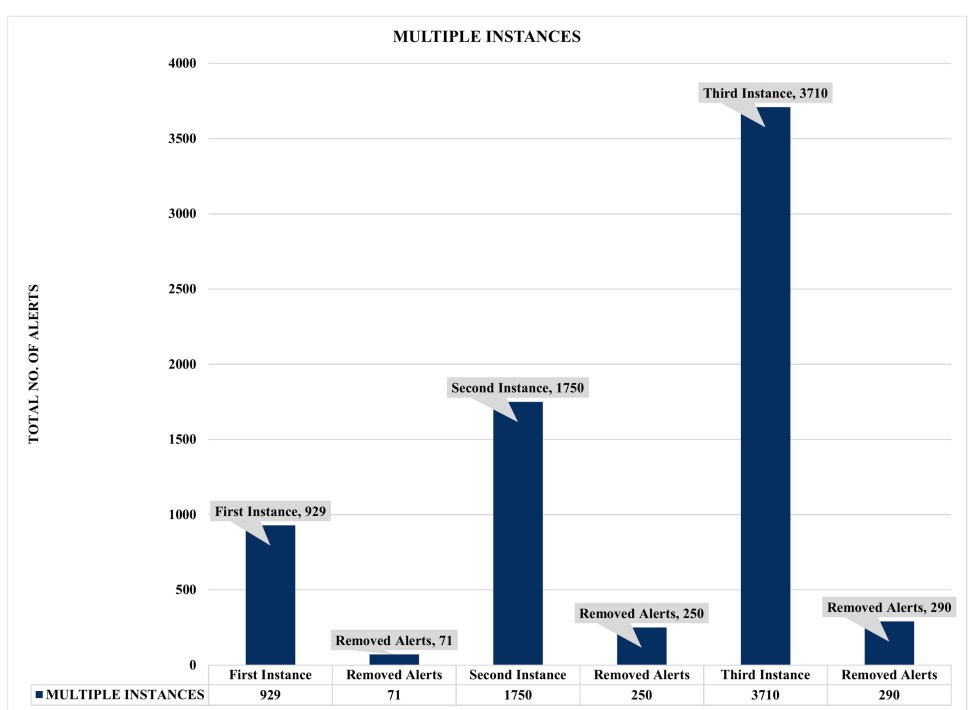

**Figure 11** **Assessment of multiple instances alert aggregation and correlation.**

the authors believed this is an excellent performance considering the dataset sample. Also, it will be fair to state that processing these irrelevant alerts will waste valuable computational and human resources. Similarly, Fig. 11 shows the findings of multiple instances ranging from 1,000 to 4000 alerts as the sample sizes for the individual evaluations. Again, the results demonstrated that SEC had achieved a decent correlation performance on the various sets.

For instance, Fig. 11 shows that out of 4000 alerts, SEC efficiently and effectively compressed it to only 3710 alerts by removing a whopping 290 irrelevant alerts that could have unnecessarily exhausted an organization's valuable resources, such as human and computational resources. Likewise, SEC replicates a similar performance for the 2000 and 1000 alert samples, eliminating a vast 250 and 71 irrelevant alerts for both instances, respectively. Therefore, considering the above significant performance of SEC for both single and multiple instances, we can argue that the proposed method could significantly contribute to the solutions of analyzing and managing the massive irrelevant alerts generated by current IDSs.

## Assessment of single instance with multiple correlation time (COTIME)

This section presents numerous experiments based on a single number of inputs with varying correlation times (COTIME(s)). It also further assess the performance and consistency of the proposed security event correlator based on alert prioritization that efficiently demonstrated the number of high, medium, and low priority alerts. Furthermore,

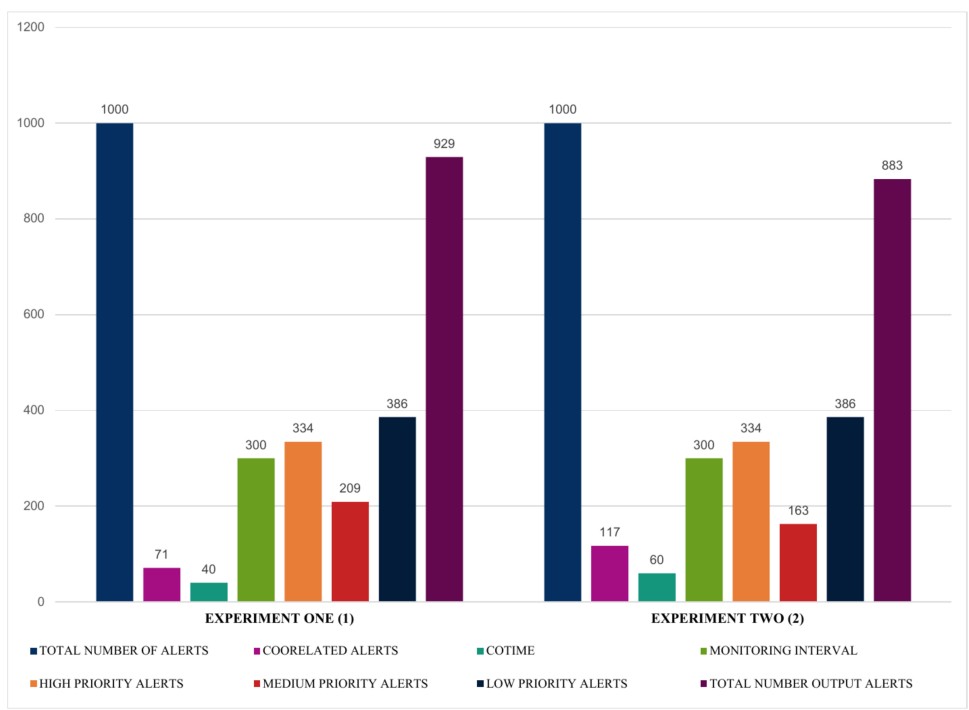

**Figure 12** Illustration of the first two experiments of alert prioritization with multiple COTIME.

it shows the correlation time and the correlated alerts using a fixed sample size of 1000 alerts.

The findings presented in Figs. 12 and 13 summarized the evaluation results obtained from four experiments, highlighting some exciting results. For instance, experiments one and two illustrated in Fig. 12 show that using a fixed sample of 1000 alerts and a COTIME of 40 s, SEC efficiently identified 71 alerts as correlated alerts, producing a manageable 929 alerts as final relevant alerts. Similarly, out of the 929 alerts, SEC effectively categorized a considerable 334 alerts as high priority alerts, 209 alerts as medium alerts, and 386 low priority alerts. Likewise, experiment two presented in Fig. 12 indicates that increasing the number of COTIME to 60 s results in even better findings, such as identifying a massive 117 alerts as correlated alerts, thereby leading to only 883 alerts as the final output alerts. Unlike experiment one, only 163 alerts were categorized as medium priority alerts, while 334 and 386 were recorded for high and low alerts, respectively.

Moreover, Fig. 13 further validates that an increase in COTIME is a crucial factor in the performance of the SEC. For example, increasing the COTIME to 90 and 120 s results in more correlated alerts like 124 and 128 alerts for 90 and 120 s, respectively. Also, experiments three and four have effectively identified 157 and 154 alerts as medium alerts. The final output alerts for the two experiments are 876 and 872 alerts, respectively, with a low priority alert of 385. However, the values for high priority alerts for all these four

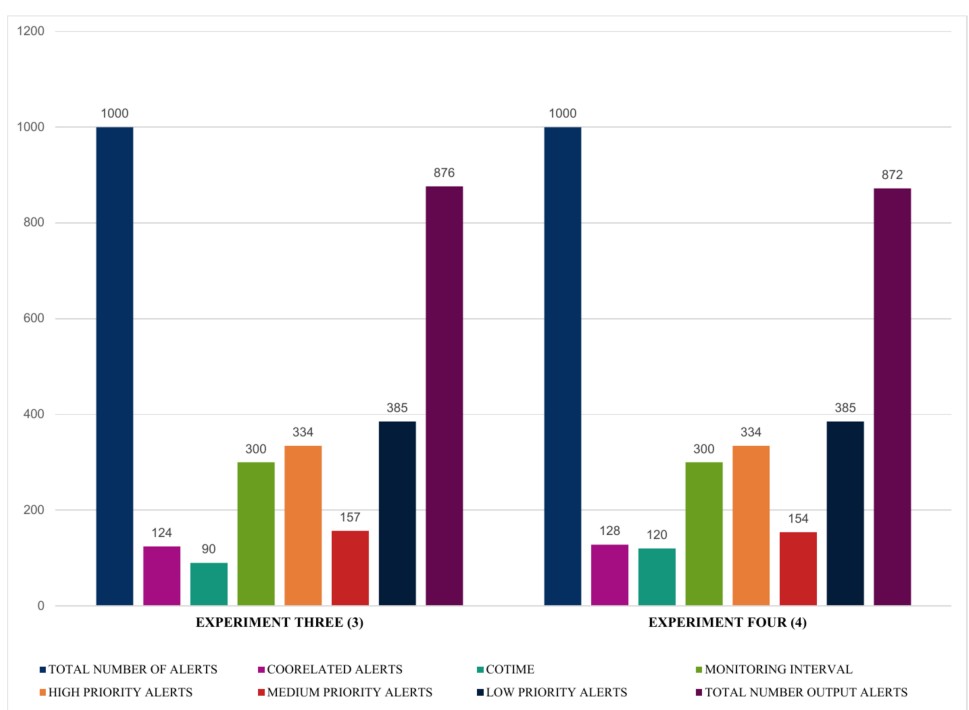

**Figure 13** Demonstration of single instances with multiple COTIME for alert prioritization.

experiments presented in Figs. 12 and 13 recorded the same values. Nonetheless, we have conducted a series of experiments to validate these similarities.

Similarly, the results illustrated in Figs. 14 and 15 shows that SEC obtained impressive findings such as an increase of 133 and 165 correlated alerts for experiments five and six with a COTIME of 180 and 300 s, respectively. Again, this further validates that COTIME significantly correlates with the number of correlated outputs. Moreover, SEC categorized 150 and 131 alerts as medium alerts, while 383 and 374 alerts were low priority alerts with only 867 and 835 final output alerts for experiments five and six. Nevertheless, like the previous experiments, the values for high priority remain as 334 alerts for both experiments, as shown in Fig. 14.

Finally, and most importantly, Fig. 15 presents some notable and exciting findings that reveal interesting correlations among the chosen factors of the conducted experiments. For instance, experiment seven obtained a massive 474 correlated alerts due to a 600 s increase of COTIME. As a result, it resulted in a much proportionate distribution of alerts into various categories like 223 high priority alerts, 227 low priority alerts, a negligible 78 medium alerts with only 526 final output alerts. Similarly, SEC achieves exciting results when COTIME is 900 s, such as a manageable final output of 424 alerts dues to the vast 576 correlated alerts. Also, experiment eight presented in Fig. 15 effectively and efficiently categorized the final output into 178, 65, and 181 alerts for high, medium, and low priority, respectively. As a result, it would be fair to conclude that SEC can significantly assist the network or system administrators with a considerably simplified analysis of alerts. Moreover, based

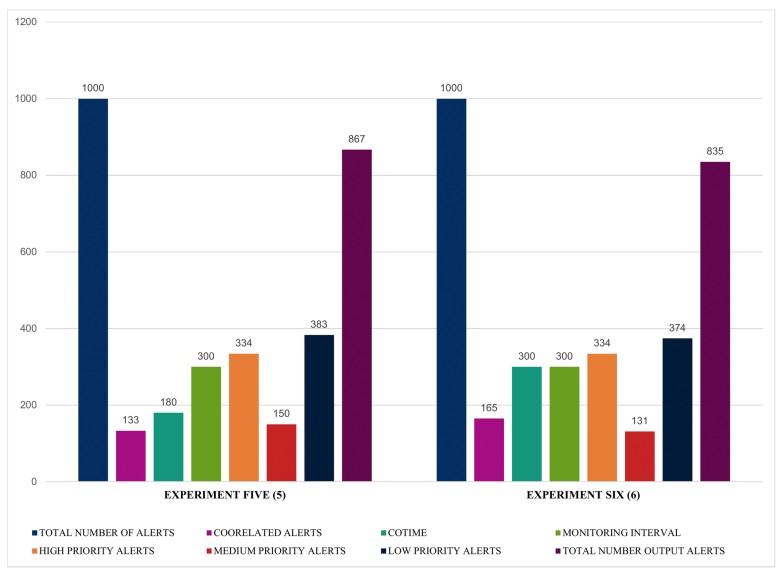

**Figure 14** Further illustration of alert prioritization of single instances with multiple COTIME.

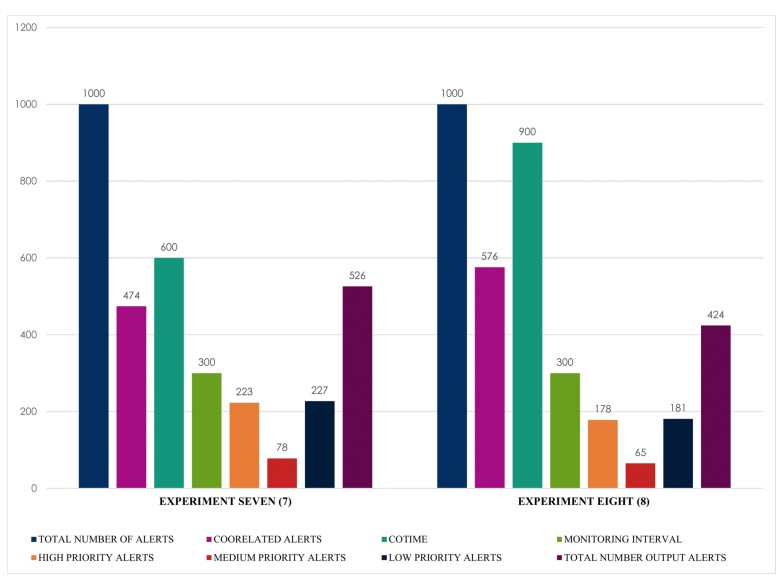

**Figure 15** Alert prioritization for single instances with multiple correlation time (COTIME).

on the results presented above, it is self-evident that while COTIME increases, the number of output alerts decreases. Therefore, we can conclude that the number of correlated alerts and COTIME are directly proportional to each other.

## Assessment of the time factor on alert correlations

Similarly, this section meticulously conducted more experiments to evaluate the correlation of time factors and the performance of the proposed approach (SEC). Considering that the

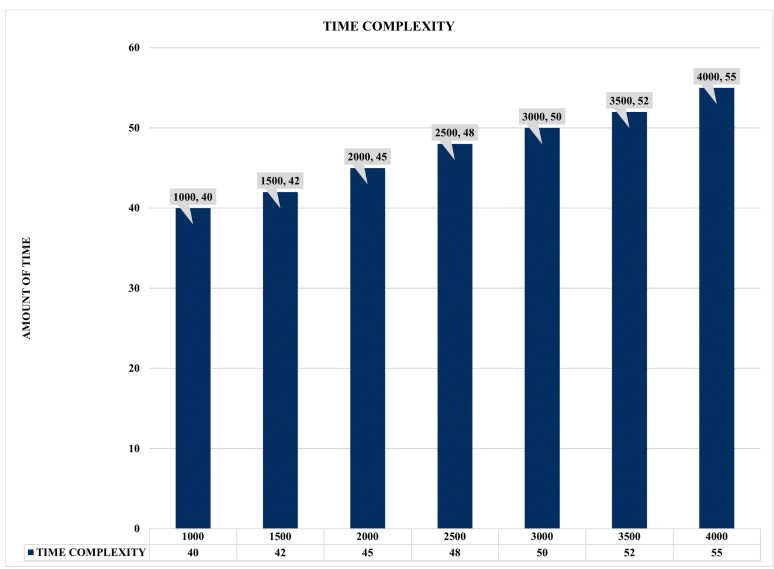

**Figure 16  Time complexity graph with multiple instances.**

amount of COTIME has significantly influenced the outcomes of alert correlation achieved by SEC, this section intends to validate this apparent relationship.

The results illustrated in Figs. 16 and 17 present the evaluation performance of correlation time against the correlated alerts with some interesting findings. For instance, Fig. 16 shows the time lag or time complexity on static COTIME with multiple instances ranging from 1,000 to 4000 alerts. Moreover, time complexity or lag is the time difference between various sample inputs. Similarly, Fig. 16 confirms a continuous increase in the time (COTIME) as the value of the inputs increases, which validates the results presented in Figs. 13, 14, and 15. For instance, increasing the inputs to 1500, 2000, and 2500 alerts has increased COTIME to 42, 45, and 48 s, respectively. Likewise, the sample of 3500 and 4000 alerts recorded 52 and 55 s of COTIME, respectively. Based on these performances, the authors have concluded that COTIME is a crucial factor in SEC's better performance, as shown in Fig. 16 and preceding evaluations.

Moreover, Fig. 17 shows a more compelling assessment of COTIME with correlated alerts. For instance, there is a continuous rise in correlation time and the correlated alerts, like it takes 40 s COTIME to correlate 71 correlated alerts. Also, the increase of COTIME to 90, 120, and 182 s has achieved 117, 124, and 128 correlated alerts, respectively. Furthermore, the considerable rise of COTIME to 300, 600, and 900 s has achieved some interesting findings such as 165, 474, and 576 correlated alerts, which is a significant and impressive performance for SEC. Based on the above results, it will be fair to conclude that a massive increase in COTIME could lead to reliable and effective alert correlation. However, this could also lead to the demand for more processing power and other similar burdens. Irrespective of these challenges, SEC has achieved its objective of significantly minimizing the massive irrelevant alerts generated by heterogeneous IDSs. Also, SEC has

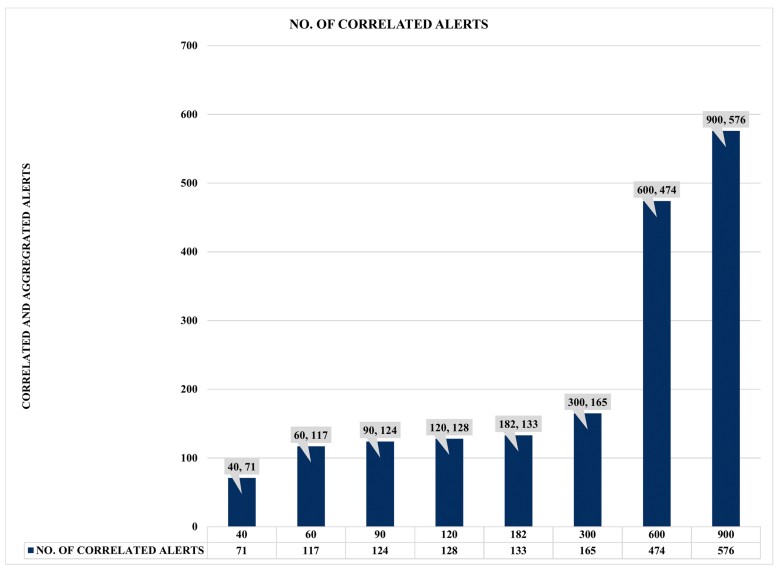

**Figure 17** Comparison of correlation time (COTIME) with correlated alerts.

enabled easy analysis of alerts, which has always been a considerable challenge for system and network administrators.

## CONCLUSIONS

Data security has been a massive concern over the past decades due to the considerable high-tech progression that has positively influenced human society in many aspects. However, the illegal mining of data due to the vulnerabilities of security mechanisms has enabled malicious users to compromise and exploit the integrity of existing systems, thereby causing colossal havoc to individuals, governments, and even private sectors. Consequently, this leads to the necessity to deliver reliable and effective security mechanisms by harnessing various techniques to design, develop and deploy optimized IDSs, for example, Snort-based IDS. Nonetheless, existing studies have shown that manually creating Snort rules, which is highly stressful, costly, and error-prone, remain a challenge.

Therefore, this paper proposed a practical and inclusive approach comprising a Snort Automatic Rule Generator and a Security Event Correlator, abbreviated SARG-SEC. Firstly, this paper provides a solid and sound theoretical background for both Snort and alert correlation concepts to enlighten the readership with the essential idea of understanding the presented research content. Additionally, this paper presents an efficient and reliable approach (SARG) to augment the success of Snort, which significantly minimizes the stress of manually creating Snort rules. Moreover, SARG utilizes the contents of various pcap files as live attacks to automatically generate optimized and effective Snort rules that meet the entire criteria of Snort rule syntaxes as described in existing literature (*Khurat & Sawangphol, 2019*). The results presented in this paper have achieved impressive performances in the auto-generation of Snort rules, with little knowledge of how the

contents of the rules are generated. Furthermore, the auto-generated Snort rules could serve as a beginning point for turning Snort into a content defense method that considerably lessens data leakages.

Moreover, this paper posits an optimized and consistent Security Event Correlator (SEC) that considerably alleviates the current massive challenges of managing the immense alerts engendered by heterogeneous IDSs. This paper evaluated SEC based on single and multiple instances of raw alerts with consistent and impressive results. For example, utilizing a single sample of 1000 alerts and multiple instances of 1000 to 400 alerts, SEC effectively and efficiently identified 71 and 290 alerts as correlated alerts. Furthermore, this paper uses a single instance of 1000 alerts with varying COTIME to further measure the performance and stability of SEC based on alert prioritization that competently revealed the number of high, medium, and low priority alerts. For example, SEC achieved excellent performances on a single instance of 1000 alerts like 133 and 165 correlated alerts for experiments five and six with a COTIME of 180 and 300 s, respectively. Furthermore, SEC identified 383 and 374 low priority alerts, whereas 150 and 131 alerts are medium alerts with only 867 and 835 final output alerts for experiments five and six.

Lastly, to further confirm the significant correlation between the number of correlated outputs and COTIME. Experiments seven and eight present exciting results that demonstrated some intuitive relationships amongst these chosen factors. For instance, an increase of COTIME to 600 and 900 s shows a much balanced and acceptable alert prioritization into numerous categories like final output alerts of only 526, 227 low priority alerts, 223 high priority alerts, and a negligible 78 medium alerts. Based on the above findings, it would be fair to conclude that SARG-SEC can considerably support or serve as a more simplified alert analysis framework for the network or system administrators. Also, it can serve as an efficient tool for auto-generating Snort rules. As a result, SARG-SEC could considerably alleviate the current challenges of managing the vast generated alerts and the manual creation of Snort rules.

However, notwithstanding the decent performance of the SARG-SEC, it has some apparent downsides that still necessitate some enhancements. For instance, the auto-generation of consistent *sid* values of the respective rules and why SARG has two contents in a single generated Snort rule. Similarly, we acknowledged that the sample sizes of alerts and the apparent relationship of correlated alerts with COTIME could challenge our findings. Nevertheless, we aim to extend this research to establish means of solving these research challenges. In the future, we intend to: (i) Extend SARG's functionality to efficiently generate consistent *sid* values for the auto-generated Snort rules and establish concepts of how SARG automatically generated the contents of the rules. (ii) Investigate the apparent relationship of COTIME and correlated alerts and present solutions to how we can correlate a larger sample size of alerts within an acceptable time frame to mitigate the need for unceasing computing resources. (iii) Finally, evaluate SEC within a live network environment instead of pcap files and auto-generate reliable and efficient Snort rules using the knowledge of the proposed anomaly IDS (*Jaw & Wang, 2021*).

## ACKNOWLEDGEMENTS

Special thanks to Professor Wang Xue Ming for the guidance and supervision. Similarly, the authors are grateful to the editor and reviewers for their valuable comments and suggestions. Lastly, Sherriffo Ceesay, and Eva Lee Redding, thanks for the insightful comments, motivation, and advice.

### Funding
The authors received no funding for this work.

### Competing Interests
The authors declare there are no competing interests.

### Author Contributions
- Ebrima Jaw conceived and designed the experiments, performed the experiments, analyzed the data, performed the computation work, prepared figures and/or tables, authored or reviewed drafts of the paper, and approved the final draft.
- Xueming Wang conceived and designed the experiments, performed the experiments, analyzed the data, performed the computation work, authored or reviewed drafts of the paper, and approved the final draft.

### Data Availability
The raw data is available at Github: https://github.com/KhadimJr/SARG-SEC.git.

### Supplemental Information
Supplemental information for this article can be found online at http://dx.doi.org/10.7717/peerj-cs.900#supplemental-information.

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
