# Peer review of "A novel hybrid-based approach of snort automatic rule generator and security event correlation (SARG-SEC)"

_PeerJ Computer Science, doi:10.7717/peerj-cs.900_

## Round 0.1 · original submission · Major Revisions

Kindly improve the language and presentation of this manuscript.

Also, Reviewer 3 has requested that you cite specific references. You may add them if you believe they are especially relevant. However, I do not expect you to include these citations, and if you do not include them, this will not influence my decision.

Reviewer 1 ·

Basic reporting

The article uses clear, unambiguous, technically correct text, having a good introduction clearly mentioning the disadvantages of the manual snort rules generation. However, the paper is a little bit long and wordy and in some sections is exaggerated.
Literature is not sufficient to show the disadvantages of existing automatic snort reules generation and the need for current work that fills the gap. For example, see https://doi.org/10.1109/IranianCEE.2016.7585840 and https://doi.org/10.1109/ICITEED.2015.7409013 work is also related to this, many more example exists.
The structure of the paper is not bad, the figures and tables look fine.
The results partly support the hypothesis proposed by the authors.

Experimental design

With much detail design of the experiment is outlined in order to evaluate the proposed snort automatic rule generator and security event correlation. However, there is no proper explanation and discussion of attack types under which SARG-SEC is evaluated. Attack type is needed to explain under which the proposed strategy is evaluated.

Validity of the findings

In the abstract from lines 50-52” It is evident from the experimental results that SARG-SEC has demonstrated impressive performance and can provide competitive advantages compared to other related approaches”. But not linked with a conclusion as there is a comparison shown with other related approaches.

The author needs to show the comparison of manual rules results Vs auto-generated snort rules results.

Reviewer 2 ·

Basic reporting

How to define the snort rules is one the critical issue when deploying snort-based IDS.
This paper made a good trial to automatically generating the rule for such IDSs, which I think is an interesting,challenging and meaningful trial.

Experimental design

no comment

Validity of the findings

no comment

Reviewer 3 ·

Basic reporting

The authors present a novel hybrid-based approach of snort automatic rule generator and security event correlation (SARG-SEC). In general, the work is interesting and the proposed methodology is adequately explained. The authors rely on Snort, a well known Network Intrusion Detection System (NIDS). Although the contributions of the paper are mentioned in the introductory part, is not very clear how the paper is differentiated with respect to other works. Moreover, the introductory part can be further enhanced, including some statistics about NIDS and real cybersecurity incidents. The authors provide a detailed background on NIDS and Snort. Finally, despite the fact that the evaluation results demonstrate the efficiency behind this work some further clarification should be provided

Experimental design

Although the authors present a detailed overview about NIDS and Snort, the paper does not include a paper about similar works. Some indicative references are given below. Moreover, it is not very clear why the authors chose Snort? why not other signature-based NIDS like Suricata.

[1] Radoglou-Grammatikis, Panagiotis, et al. "SPEAR SIEM: A Security Information and Event Management system for the Smart Grid." Computer Networks 193 (2021): 108008.

[2] Grammatikis, Panagiotis Radoglou, et al. "SDN-Based Resilient Smart Grid: The SDN-microSENSE Architecture." Digital 1.4 (2021): 173-187.

[3] Grammatikis, Panagiotis Radoglou, et al. "Secure and private smart grid: The spear architecture." 2020 6th IEEE Conference on Network Softwarization (NetSoft). IEEE, 2020.

[4] Sekharan, S. Sandeep, and Kamalanathan Kandasamy. "Profiling SIEM tools and correlation engines for security analytics." 2017 International Conference on Wireless Communications, Signal Processing and Networking (WiSPNET). IEEE, 2017.

[5] Suarez-Tangil, Guillermo, et al. "Providing SIEM systems with self-adaptation." Information Fusion 21 (2015): 145-158.

[6] Suarez-Tangil, Guillermo, et al. "Automatic rule generation based on genetic programming for event correlation." Computational Intelligence in Security for Information Systems. Springer, Berlin, Heidelberg, 2009. 127-134.

[7] Kotenko, Igor, et al. "Parallelization of security event correlation based on accounting of event type links." 2018 26th Euromicro International Conference on Parallel, Distributed and Network-based Processing (PDP). IEEE, 2018.

[8] Fedorchenko, Andrey, and Igor Kotenko. "IOT Security event correlation based on the analysis of event types." Dependable IoT for Human and Industry: Modeling, Architecting, Implementation 147 (2018).

[9] Ferebee, Denise, et al. "Security visualization: Cyber security storm map and event correlation." 2011 IEEE Symposium on Computational Intelligence in Cyber Security (CICS). IEEE, 2011.

[10] Dwivedi, Neelam, and Aprna Tripathi. "Event correlation for intrusion detection systems." 2015 IEEE International Conference on Computational Intelligence & Communication Technology. IEEE, 2015.

Validity of the findings

Although the authors provide a lot of figures related to the effectiveness of the proposed methods, it is not clear how the authors reflect the validity of the findings. For instance, the authors can compare their implementation with other works, other IDS and Security Information and Event Management (SIEM) systems, using some baseline metrics.

Additional comments

The paper should be re-checked entirely about potential writing errors and typos.

·

Basic reporting

Spelling mistakes such as the one in line 594 should be avoided where "A priori" is misspelt as "Apriorior". In addition, The grammar within this article needs another round of proofreading. In line 176, instead of just "propose", one could use "to propose". In the same line, the conjunction doesn't really do justice to an intuitive reading of the line and feels awkward. Multiple such issues were observed all over the manuscript.
The figures and graphs look somewhat low in quality (both in terms of resolution and the font used). It's advised to export these in 300 dpi for publishable quality.

Experimental design

no comment

Validity of the findings

no comment

---

## Round 0.2 · Minor Revisions

I understand you wish to cut down the length of your manuscript. Feel free to do so, and kindly carefully proofread the paper during your process.

Reviewer 3 ·

Basic reporting

The authors addressed all the comments. Therefore, the paper is accepted for publication

Experimental design

The authors addressed all the comments. Therefore, the paper is accepted for publication

Validity of the findings

The authors addressed all the comments. Therefore, the paper is accepted for publication

Additional comments

The authors addressed all the comments. Therefore, the paper is accepted for publication

·

Basic reporting

Looks good

Experimental design

Looks good

Validity of the findings

Looks good

---

## Round 0.3 · accepted · Accept

Thank you for providing an improved version of your manuscript. This manuscript looks good for acceptance.